# August Vollmer, Traffic in Women, and the Theory of Organized Crime

Paul Knepper 

Department of Justice Studies, San José State University, San Jose, CA 95192, USA; paul.knepper@sjsu.edu

**Abstract:** During the 1920s, the League of Nations carried out the first intercontinental investigation into the traffic in women. Although this work is virtually unknown in criminology, the investigators, William Snow and Bascom Johnson, formulated the conceptual language of "trafficking" used today. It was also during the 1920s that Frederick Thrasher and John Landesco published their pioneering works on "organized crime" drawing on research in Chicago. The advantages of the League's model can be seen in the response to a 1924 report of a white slave traffic ring in Los Angeles by August Vollmer, the celebrated founder of professionalism in American policing. Vollmer's language of a white slave traffic ring in Los Angeles recalls a nineteenth-century understanding of traffic in women but previews the illegal enterprise model that emerges from the industrial city. Drawing on their understanding of crime in port cities, Snow and Johnson situate the traffic in women within a social networks model. Vollmer looked for the spider, Snow and Johnson looked at the web.

**Keywords:** human trafficking; organized crime; illegal enterprise theory; criminal network; transnational crime; historical criminology



## 1. Introduction

During the 1920s, the League of Nations conducted the first international investigation into the traffic in women. While exploring the situation in California, the League's investigator received a startling account of a "white slave traffic ring" operating in Los Angeles. White slavers brought women from France, Italy, and Mexico, working within a secret criminal organization that included smugglers and gamblers but also leading public officials throughout the city. The ring included the presiding judge of the women's court, the mayor's chief of staff, as well as an appellate judge and a U.S. marshal. It was a fantastic story, comparable to the script of a Hollywood movie, and as the investigator explained, "Were the information coming from any source other than Chief Vollmer, its credibility might be questioned" (Worthington 1925).

August Vollmer became known as the founder of American policing for innovations he introduced to the Berkeley Police Department from 1906 to 1932. Vollmer's contribution to criminology has been assessed from the standpoint of advancing police professionalism (Wilson 1953; Parker 1961; Douthit 1975; Carte and Carte 1976; Oliver 2017). During 1923–1924, Vollmer left Berkeley to serve as chief of the Los Angeles Police Department. His report of a secret conspiracy of international criminals in the city is interesting because in the decades before the First World War, investigations in New York, Chicago, and elsewhere determined gangs of white slavers to be more myth than reality. Investigations in the 1920s, including that of the League of Nations, arrived at this same conclusion. Historians refer to the white slavery campaign as a "panic" or "scare" (Walkowitz 1992; Donovan 2006; Slater 2007; Keire 2010).

Even more interesting is what Vollmer's report, and the alternative to it by the League's investigators, contributes to the theory of organized crime (Kleemans 2014). During the 1920s, significant work took place in understanding organized crime, including path-breaking studies by Thrasher (1927) and Landesco (1929). This effort perceived organized

crime in the context of the industrial city, emphasizing local markets for illegal products, provision of illicit services, and manipulation of criminal justice proceedings. It led to the illegal enterprise theory of organized crime (Haller 1990; Liddick 1999; Reuter 1983). The League's investigators, William Snow and Bascom Johnson, pursued an understanding of commercial sex that extended across continents, responding to economic supply and demand in a worldwide political economy. They discounted Vollmer's report because it did not fit the theory of organized crime they had developed. They abandoned the concepts of "rings" and "syndicates", and formulated instead the concept of "trafficking" (Chin and Finckenauer 2011). Their work supports the social networks approach to organized crime (Carrington 2011; Finckenauer and Chin 2011; Kleemans and van de Bunt 2008; Morselli 2009).

The research presented here uses the methods of historical criminology (Churchill 2017; Churchill et al. 2022; Knepper 2014; Knepper and Scicluna). Historical criminology can be defined as "research which incorporates historical primary sources while addressing present-day debates and practices in the criminal justice field" (Lawrence 2019, p. 495). The primary sources for the investigation into the traffic in women by the League of Nations are found in the League of Nations Archives at the United Nations Library in Geneva. The collection includes Vollmer's account of trafficking in Los Angeles as well as field reports and related documents from the three-year investigation led by Snow and Johnson (Knepper 2012, 2016). Historical criminology emphasizes the importance of time. It is not committed to a view of history as a singular, linear sequence of events that can be divided into discrete periods or eras, but rather historical time as a plural or multi-layered experience, marked by abrupt transitions, gradual change, and stable continuities (Churchill 2017). In reading archival documents, the 1920s emerges, not as a specific historical era—the Interwar Period or Jazz Age—marked by unique historical events, but as an historical experience that continues into the present. This long-term view is more useful for theorizing than the "newest" data.

This discussion is divided into three parts. The first part explains what the League of Nations was doing in California. The second part examines Vollmer's report of white slave trafficking in Los Angeles. The third part develops trafficking as a theory of organized crime.

## 2. The League of Nations in California

At the end of the First World War, the Allies established the League of Nations. Although the League failed as a political institution, it had a remarkable role in public health, women's issues, child welfare, refugee settlement, and other social issues (Garcia et al. 2016). This included crime and criminal justice. The Social Section worked with national governments and international voluntary organizations such as the International Criminal Police Commission and International Penitentiary Commission. The League mapped illicit drug smuggling, curtailed counterfeiting of currency, agreed on standards for prison conditions, formulated a working definition of political terrorism, and conducted the first international inquiry into the traffic in women (Block 1989; Knepper 2012; Petruccelli 2016; Kozma 2017).

When the Constitution for the League was signed, it included the provision, Article 23c, that the League would assume responsibility for treaties concerning the white slave traffic agreed before the war. In 1921, representatives from 34 nations convened in Geneva to assess the level of compliance with agreements reached in Paris in 1904 and Madrid in 1910. This conference replaced the language of white slavery with that of trafficking. Article 13, known as the Final Act, recommended the phrase "white slave trade" should be replaced in international agreements with that of "traffic in women and children" (League of Nations 1921, p. 600). This provision avoided the racist implication that only European women mattered, and although in practice the League continued to focus its efforts on European women as if they were the only women who mattered (Schettini 2017), the new language signaled the League's ambition of becoming more international than European. In

adopting the language of trafficking, the League of Nations also situated the exploitation of women within an awareness of global organized crime. H. Wilson Harris, who popularized the League's traffic in women investigation in his 1928 book *Human Merchandise*, realized the wider significance of the change in words. The language in Article 23c of the League Constitution had perhaps unintentionally linked traffic in women with traffic in drugs, but the course of the League's activities demonstrated "a common terminology applied to both evils—traffic in drugs, traffic in women" (Harris 1928, p. 26).

The 1921 Geneva conference also provided for the establishment of an inter-governmental agency to suppress the cross-border trade in women's bodies. The Advisory Committee on the Traffic in Women, comprised of representatives from member states, became a technical organization within the League framework. Rachel Crowdy, who led the committee, was chief of the League's Social Section. She was the only woman to lead an agency within the League of Nations, and this was appropriate given that women led the international campaign against the traffic in women (Gorman 2012; Limoncelli 2010). It is also significant in this context because women's leadership enabled the Advisory Committee to pursue an understanding of trafficking. The International Criminal Police Commission (ICPC)—becoming known as Interpol in 1956—had, from its founding in 1923, aspired to become a technical organization within the League framework. For nearly a decade, ICPC leader Johannes Schober lobbied Geneva for recognition. Although Crowdy permitted an ICPC member to join sessions of the Advisory Committee on the Traffic in Women as an unofficial member, she successfully resisted further cooperation because she did not want the "police view" to dominate the Social Section.

The United States never joined the League of Nations, yet Americans charted the course of the anti-trafficking campaign. The Advisory Committee invited the U.S. government to send an unofficial representative and the Harding Administration named Grace Abbott, head of the Children's Bureau in Washington D.C. (Everett 1927). In 1923, at the Advisory Committee's second meeting, Abbott proposed a worldwide investigation into the traffic in women. The investigation would determine whether there was an international traffic in women for prostitution, learn the routes and methods of traffickers, and judge the effectiveness of national measures to suppress it. Above all, the study was necessary to secure the facts of trafficking needed to refute sensational exaggerations in the press and general denials from government authorities (League of Nations 1923, p. 27). The Advisory Committee welcomed Abbott's proposal, and the League Council agreed on the condition the investigation would be financed with outside funds. When Abbott acquired $75,000 from the Bureau of Social Hygiene in New York, the League approved the project. The Bureau of Social Hygiene had been established by John D. Rockefeller Jr. to promote research into public health and social problems. Rockefeller Jr, one of the wealthiest persons in the world at the time, had taken a reformer's interest in prostitution since 1909 when he chaired a grand jury investigation into white slavery in New York City (Knepper 2016).

To oversee the investigation, the Advisory Committee assembled a Special Body of Experts from Italy, France, Britain, Belgium, Uruguay, Switzerland, Japan, and the United States. The role of Abbott and Rockefeller meant that the United States would control the investigation and the Council agreed to appoint Dr. William F. Snow to chair the experts. Snow was director of the American Social Hygiene Association (ASHA), an organization funded by Rockefeller's Bureau of Social Hygiene. To manage the field investigation, Snow chose Bascom Johnson, the ASHA's director of Legal Services. Snow, a medical doctor, and Johnson, a lawyer, had served together in the U.S. Army during the First World War. In their work for the public health section, they tried to shield soldiers from venereal disease by keeping prostitutes away from training camps. After the war, Colonel Snow joined the faculty of public health at Stanford University, and Major Johnson led the ASHA's legal campaign for the abolition of prostitution (Hennigan 2004; Keire 2010).

Snow and Johnson modelled the League's investigation on the method Johnson and other leaders of the ASHA had used to close red-light districts. They helped local reformers in San Francisco, New York, Chicago, and other cities set up municipal vice commissions,

find undercover investigators, and implement reforms. The ASHA took the view that "commercial vice", on the scale that took place in red-light districts, could not occur without the connivance of the police and local authorities. So, they taught the vice commissions to use undercover investigations. The investigators gathered first-hand information about arrangements by police and local authorities to profit from prostitution, gambling, and other vice activities. The vice commissions would then meet with city officials to urge reform. The reformers would confront them with extensive documentation of corruption, and if city leaders resisted, threaten to place their information before the public (Fronc 2009; Keire 2010).

To conduct the League's investigation, the Special Body of Experts distributed a survey to national governments and collected reports from international voluntary associations. They also conducted "on-the-spot" or field investigations. Johnson, and to some extent, Snow, carried out interviews in various cities: they spoke with 1500 police, health, immigration, and other officials. For the most part, they relied on undercover interviews. Johnson supervised eight individuals who produced some 5000 interviews with prostitutes, brothel-keepers, pimps, and others involved in the sex trade. The undercover operatives included individuals of various backgrounds from several countries, and included some women, although most field reports were done by American men trained by the ASHA. For three years, 1924, 1925, and 1926, the League conducted field investigations in 112 cities in 28 countries across the Americas, Europe, and the Mediterranean (League of Nations 1927a; Everett 1927).

George Worthington carried out the League's field investigations in California in 1924. A lawyer who worked for the ASHA's legal section, and later became director of the Committee of Fourteen in New York City, he also investigated the situation in Canada, Cuba, and United States. In Los Angeles, Worthington visited the Police Department. Captain R. Lee Heath, who had replaced Vollmer in August 1924 as chief, sent him to Lieutenant W.M. Littell, assistant commander of the vice squad. Littell declared there were no more open houses of prostitution in the city, very little street-walking, and "No evidence whatsoever of white slavery nor international traffic" (Worthington 1924a). John H. Pelletier, Executive Secretary of the Morals Efficiency Association, affirmed Littell's conclusion. There were no more commercialized houses of prostitution. Prostitution took place in hotels, rooming houses, apartments, and automobiles. "He", Worthington wrote of Pelletier, "has no evidence regarding white slavery, nor international traffic; doesn't believe there is any in Los Angeles" (Worthington 1924a). Worthington's interview with Walter E. Carr, Inspector in Charge, U.S. Immigration Service in Los Angeles yielded the same result: "there is little white slavery in Los Angeles" (Worthington 1924a).

In San Diego, Worthington talked to Captain P.J. Hayes, in charge of the detectives. Hayes had "found no evidence of International White Slavery in ten years". He had investigated complaints and found them to be groundless (Worthington 1924b). In San Francisco, Worthington met Captain Duncan Matheson who led the detectives. Matheson said "there has been no white-slave traffic since the closing of the Tenderloin" and that the "pimps who used to hang out there had gone to Buenos Aires or Montevideo". T.G. Doser, the Inspector in Charge, Immigration Service, San Francisco said there was "no evidence at all of white slavery, but that there was some 'Yellow' slavery, i.e., Chinese girls—among the Chinese, but not the whites". There had been no cases of traffic involving white people since the war (Worthington 1924c).

Less than a year later, Worthington learned about Chief Vollmer's account of the Los Angeles white slave ring.

## 3. Vollmer in Los Angeles

In 1925, August Vollmer had already achieved most of what would make him famous. As Berkeley's first chief of police, he organized police records and a modus operandi system, began using intelligence tests for recruiting, and introduced motorcycles and automobiles to patrol. In conjunction with the University of California at Berkeley, he

oversaw the development of the lie detector, brought forensic science to the investigation, and established a school of criminology. He served as president of the California State Chiefs of Police Association and the International Association of Chiefs of Police (Parker 1961; Douthit 1975; Macnamara 1977; Carte and Carte 1976). Raymond Fosdick celebrated Vollmer's accomplishments at Berkeley in his 1920 book *American Police Systems* (Fosdick 1920). In 1924, *Collier's Weekly*, a popular magazine, published an article about Vollmer's struggle in Los Angeles under the title "A Professor Who Cleaned Up a City". The article spotlighted Vollmer's accomplishments on a national stage (Collins 1924).

There was, according to Vollmer's sources, an "organized ring of white slavers in Los Angeles" that operated from a restaurant at 411 N. Main Street. The ring, led by a Frenchman and an Italian, supplied French, Italian, and Mexican women to an inter-city chain of brothels. The ring worked the women in one location for a brief period, then shifted them to another. Each woman travelled with a "passport", communicated in code, to men working for the ring at each site. In San Diego, the ring's man coordinated the local network with the support of the U.S. marshal. Other members of the ring included police, newspapermen, and government authorities who contributed to a "syndicate" of betting and bootlegging activities. Vollmer had tried to make arrests but found himself blocked in state and federal courts by invisible forces (Worthington 1925).

Vollmer named members of the ring. Some, such as "Chito", "Domingo", and "Ernie", remain in the shadows. Others, including Albert Marco, Guy McAfee, and Kent Kane Parrot would have made a "Who's Who" list of notorious persons in 1920s Los Angeles. All three worked with Charles Crawford, referred to by the *Evening Express* as the "city's outstanding underworld boss". When he was shot to death in 1927, city newspapers exposed his activities and associates (Rayner 2009, p. 87). Albert Marco came from Seattle to manage a string of brothels for Crawford, an arrangement that became public knowledge when the *Daily News* published the addresses of the establishments. He went to San Quentin for conspiring with several police officers to send a prostitute to trap a city council member after a bribe had failed (Rayner 2009, p. 82). Guy McAfee worked as a police captain in charge of the vice squad even though he owned several taverns and brothels. He married a woman who ran one of Marco's brothels and went into business with Crawford as well. A city magazine, *The Critic of Critics*, ran a story in 1931 entitled "Guy McAfee—the 'Capone' of L.A." that pictured him as an octopus with tentacles reaching into gambling, liquor, and real estate, as well as the police department and hall of justice (Rayner 2009, p. 158).

Kent Kane Parrot came to Los Angeles in 1907 to study law at the University of Southern California. He was admitted to the bar in 1909, and within a year, became a partner in a firm that included the dean of law from USC. He stayed with the firm until 1920 when he entered politics. Parrot became campaign manager, and later chief of staff, for mayor George E. Cryer. In 1924, the *Los Angeles Times* ran a series of seventeen articles about scandals in the Cryer administration and revealed Parrot as a major force behind the scenes. Commercial vice yielded more money to those protecting it than the combined salaries of all the police on patrol (Sitton 1985, p. 367).

The ring included several "respectable" individuals: George Richardson and Gavin William Craig. George S. Richardson made a fortune growing lemons in Ventura county before entering the University of Southern California to study electrical and mechanical engineering. He worked for a company that installed steam plants and the electrical department of the Los Angeles Railway company before returning to USC to study law. He was admitted to the bar in 1910 and began the practice of law. He was active in the Republican Party, and by 1915 was regarded as "one of the strong men of that organization". In 1915, he was elected as police judge in charge of the women's court (Historical Record Company 1915, pp. 847–48). Richardson, according to Vollmer, dined regularly at the restaurant on Main Street used by the white slave gang.

Gavin William Craig was born in Nebraska. He completed his law degree at the University of Southern California School of Law and was admitted to the state bar in 1901. He served as an administrator at USC and a deputy district attorney in Los Angeles County

before becoming a commissioner for Los Angeles Superior Court. He served as a judge, Los Angeles Superior Court, from 1911 to 1921, and as an associate justice, California Court of Appeal, Division 2, from 1921 to 1935. Parrot had been one of Craig's students at law school, and it was during his years as an "errand boy for Judge Craig" that he perfected his skills as a political boss. Craig had proposed Cryer in the 1921 municipal election with Parrot as chair of his campaign organization (Sitton 1985, p. 371).

What are we to make of Vollmer's account? The "white slave" language evokes the melodrama of the nineteenth-century campaign and had been rejected as inappropriate by many within the movement. As the Italian delegate to the 1902 International Conference on the White Slave Traffic pointed out, the phrase "white slave traffic" was inaccurate. The word "white" did not apply to most of the victims of sexual exploitation, who were yellow, brown, black, etc., and the "slave traffic" implied conditions of travel that seldom applied to actual conditions. Most victims had not been procured in a northern country, smuggled through a central country, nor brought to a southern country (Allain 2017, p. 7).

Nevertheless, "white slave" remained part of the American vocabulary. In 1910, the U.S. Congress passed the White Slave Traffic Act, popularly as the Mann Act after James R. Mann of Illinois who had sponsored the act in the Congress. The law made it illegal to transport women for purposes of "prostitution, debauchery or any other purpose" over international borders or across state boundaries. Enforcement of the act fell within the purview of the Bureau of Investigation—renamed the Federal Bureau of Investigation in 1935—and its director, J. Edgar Hoover, made the most of it. From 1921 to 1936, the FBI pursued some 47,500 cases. Most of these cases were "domestic cases" in the sense that they were initiated by complaints to field offices by husbands hunting run-away wives, parents seeking to prevent interracial relationships, and wives trying to recover bigamous husbands. However, Hoover continued to invoke the imagery of racial menace and sexual slavery to win publicity for the bureau's efforts. As late as 1936, Hoover's FBI claimed headlines for its war on white slavery with the prosecution of Leon Richard Smith, the so-called "colored vice overlord" (Pliley 2014).

Aside from the white slave language in Vollmer's report, there is the language of "ring" and "syndicate". A gang of white slavers in Los Angeles would have been a rare species of criminality. Investigations made into claims about the white slave trade before the war found few actual cases and no evidence of conspiracies. The Rockefeller grand jury investigation in 1909 into claims of "incorporated syndicates" and "international bands" operating in New York City turned up no evidence of any organization engaged in the traffic, and "no organized traffic in women for immoral purposes" (Rockefeller 1910). The investigation led by George Kneeland for the Chicago Vice Commission reached the same result: "It has been demonstrated that men and women engaged in the 'white slave traffic' are not organized" (Kneeland 1911, p. 41). Walter Reckless's PhD research into prostitution in Chicago to measure the impact of the Chicago Vice Commission's reforms found "no evidence in any of the cases of the existence of white slavers organized for the purpose of waylaying girls and enslaving them" (Reckless 1933, p. 42).

Vollmer's report reads similar to one of the newspaper stories about white slave gangs (De Young 1983). During the years of the League's investigation, the Advisory Committee heard a series of rumors. In 1923, the *New York Evening Post* announced the discovery of a plot to ship 500 girls to the United States via ports in Antwerp, Hamburg, and Rotterdam. The Antwerp police possessed "definite proof" that a man named Brown, who led the Girls Protection Society in Berlin, secretly operated a white slave ring. In another version, Brown planned to send 200 girls to South America aboard the *Cap Polonio* from Hamburg. Similar stories appeared about a Dutch trafficker named van Gulpen and another named Swan. Johnson and the League's investigators tracked down information across Europe and concluded that in the Brown case, journalists had sewn the narrative of an international trafficker together from bits of material pulled from police and protection society publications (Special Body of Experts 1924).

Yet it is unlikely that Vollmer had been taken in by the white slave narrative. He had first-hand information about goings-on in the city. Oliver (2017) writes that in August 1924, shortly after returning to Berkeley, Vollmer learned from the newspaper about a lawsuit against him. A woman named Charlotte Lex, who Vollmer had met in Los Angeles, filed a breach of promise suit claiming he had agreed to marry her. In the proceedings that followed, she claimed that he groped her during a visit to her house, made love to her like a "cave-man", and impregnated her in the course of seduction under the promise of marriage. Vollmer denied the accusations of sexual relations with the woman and sent investigators in search of her background. "The underworld threatened to get me shortly after my arrival in Los Angeles", he told a reporter, "and they are attempting to carry out their threats" (quoted in Oliver 2017, p. 396). Vollmer learned that Lex's attorney, Warren Williams, had ties to the underworld and had personally laid down $20,000 to force him out of his job as chief of police (Oliver 2017, p. 402). Warren Williams appears in Vollmer's report of the white slave ring. He is described as the ring's "legal representative", who "also represents professional gamblers and bootleggers by the names of Domingo, Bunco and Davis" (Worthington 1925).

Worthington said that Vollmer had first learned of the ring from a prostitute named Lillian Vail and was able to corroborate her story with information from other sources (Worthington 1925). In fact, Vollmer had conducted his own investigation of criminal activities in Los Angeles before he arrived as police chief. From 1915 to 1923, eight police chiefs had served in four city administrations, punctuated by a series of scandals. Mayors and council members had been taking campaign money and payoffs from brothel-keepers, gamblers, and liquor smugglers to look the other way. Using a discretionary fund of $100,000 provided by the Citizen's Committee, Vollmer arranged for various convicts, many on parole, to scout the terrain. He learned the names of smugglers, extortionists, and gamblers, as well as corrupt politicians and rogue police officers (Douthit 1975, pp. 108–9). He would have known that Kent Kane Parrot, who had been part of the delegation that came to Berkeley to recruit him as police chief, met regularly with Charles Crawford at the Biltmore Hotel, where Parrot had an apartment (Rayner 2009, p. 187).

Worthington acknowledged that the information had not come to him directly from Vollmer. "Mrs Van Winkle" had given him her notes from a confidential interview with Vollmer during the summer of 1925 while she was in Berkeley (Worthington 1925). Mina Van Winkle was a police lieutenant in charge of the Women's Bureau, Metropolitan Police Department, Washington, D.C., and served as president of the International Association of Police Women. She worked with Vollmer and the ASHA to establish a Crime Prevention Division within the Los Angeles Police Department (Appier 1998, p. 103). The new division consolidated the City Mother's Bureau, the Men's and Women's Parole Boards, the Juvenile Bureau, and the Women's Probation Bureau. The division coordinated police activities related to women and girls, such as the elimination of objectionable places where young girls gathered, regulation of dance halls, and stricter supervision of behaviors likely to lead juveniles into delinquency (Police Journal 1922).

Worthington styled his report on the white slave ring in Los Angeles as a "Confidential Office Memorandum" and sent it to Snow and Johnson. "Steps should be taken to follow this information up and get whatever else may be secured from Chief Vollmer" Worthington concluded (Worthington 1925). It appears that Snow and Johnson ignored Vollmer's advice. It is not clear why. If either had written a response to Worthington's memo, it is not in the archives. They certainly knew of each other's mission. Johnson led the campaign to close San Francisco's red-light district—the Barbary Coast—and actually stayed in an apartment in Berkeley for a time (Johnson 1915). In 1916, both Johnson and Vollmer spoke at a reform conference in Pacific Grove. Johnson discussed law enforcement and moral reform in northern California cities and Vollmer talked about the police and prostitution (ASHA 1916). In Canada, Snow and Johnson dispatched their ace undercover investigator, Paul Kinsie, to Montreal to follow up Worthington's report of trafficking there but not in Los Angeles.

The official *Report of the Special Body of Experts on the Traffic in Women and Children* appeared in 1927. It states "that traffic in women is extensive, although no exact estimate can be given" (League of Nations 1927a, p. 12). The main trafficking routes took place from Poland, Romania, France, and Germany to Buenos Aires, Rio de Janeiro, and Montevideo, and from the Balkans through North Africa to Alexandria and Cairo. The United States did not figure prominently in the international traffic either as a source of supply or demand. In cities such as Los Angeles, which had suddenly doubled or tripled in size, "frontier" conditions occurred. With so many young, unmarried men, conditions surrounding prostitution "remained unsatisfactory". However, "on the whole . . . there is certainly no large city and very few small ones in the United States today to which foreign women can be brought for purposes of prostitution". This was due to the absence of "open cities", which made the sex trade unprofitable, and the presence of immigration laws from 1924 made it difficult for foreign pimps to bring prostitutes into the country (League of Nations 1927b).

Snow and Johnson used information from the field reports to confirm their theory of the traffic in women. Comparing the verbatim reports from undercover investigators in the field with the official reports Snow and Johnson wrote for the Advisory Committee, it is clear that they made selective use of field information to support the recommendations they wanted to write in the reports to be published by the League of Nations. Snow and Johnson believed the United States had ended the "white slave traffic". Other governments—Argentina, France, and Egypt—should follow the American lead in abolishing legal prostitution and abandon their systems of regulated prostitution (Kozma 2017). So, whether Snow and Johnson thought Vollmer had been misinformed, Mina Van Winkle had embellished her account of what he said, or they did not have complete confidence in Worthington, did not matter. They "knew" there could be no white slave ring operating in Los Angeles because it was theoretically impossible.

## 4. The Theory of Organized Crime

The reasons why the police chief framed his account in the way it appears, and the reasons why the League's investigators overlooked it, amount to differences in theoretical interpretation. Vollmer has been characterized less as a visionary or deep thinker concerning crime in society than an innovative administrator who found technical and procedural solutions to problems in policing (Liss and Schlossman 1984, p. 81). However, his thinking about vice coincides with pioneering research into organized crime.

Neither Vollmer nor Snow and Johnson use the words "organized crime", but it is what they are writing about. The phrase appears in a Chicago Crime Commission report for 1919 but was seldom used before the 1930s (Shore 2016). Herbert Hoover became the first U.S. president to use the term when he explained the Wickersham Commission was needed to address "the growth of organized crime . . . in every part of the country" (quoted in Smith 1991, p. 137). In the 1920s, there is virtually no conceptual language to work with; academic work on the theory of organized crime had only just started. Frederic Thrasher produced the first book-length study in 1927, followed by that of John Landesco in 1929 (Finckenauer 2014). Both examined the Chicago experience within the wider interpretive framework of urban ecology championed by Ernest Burgess and Robert Park at the University of Chicago's Sociological Department (Yeager 2015; Smith 1991).

Vollmer, and Snow and Johnson, draw on their professional experience. Vollmer brought his background in policing. He conducted a detailed study of Los Angeles and formulated an action plan that would guide reforms in the future. He also learned from his consulting projects in which he conducted "surveys" of municipal police departments (including Chicago, although not until 1928). Snow and Johnson brought their experience from the campaign to close down red-light districts in various cities. Yet, in developing their understanding of organized crime, Snow and Johnson also had the advantage of information from port cities around the world that had been collected by the League's investigators. This was significant because, unlike Chicago, an industrial city that afforded racketeering and loansharking as organized crime activities, the cities Snow and Johnson

learned about facilitated an understanding of inter-continental links necessary for traffic in women and drugs. As Johnson explained, the investigators received "introductions from members of the underworld in one country to those of another" (Johnson 1928, p. 70).

Everyone—Vollmer, Thrasher, Landesco, Snow, and Johnson—refers to the "underworld" (Landesco 1934). The word has a long history in discussions of urban life and continues to appear in criminological writing about organized crime (Hartmann and von Lampe 2008; Shore 2018). Vollmer took it for granted that there was an "indolent, unscrupulous, parasitic group in every community" who shared a common desire to "live a life of ease without effort" (Vollmer 1928, p. 327). This "so-called underworld" included thieves, prostitutes, gamblers, bootleggers, drug addicts, and crooked lawyers. They were joined by "down-and-outers", persons physically or mentally incapable of living an honest existence. The underworld, though comparatively small, had a powerful influence over elections because of its willingness to pay huge sums of money to protect its interests (Vollmer 1928, pp. 326–27). This is similar to Thrasher, who said that there was, in Chicago and other large cities, an "underworld" or "criminal community", characterized by the "absence of ordinary conventions and largely given over to predatory activities and the exploitation of the baser human appetites and passions". It consisted of "human riff-raff, the hangers-on, the questionable characters, the semi-criminal classes and . . . professional criminals" (Thrasher 1927, p. 417).

In the way the word is used, it also refers to a particular social space within urban geography. Everyone mentions specific locations in cities where criminals meet to align their interests. Yet these locations represent more than conference rooms; they are pathways into the underworld as well as bridges between the underworld and the outside world, places where the invisible becomes visible. For the Los Angeles white slave ring, it is the restaurant on Main Street. For the gangsters of Chicago, it is a restaurant, cabaret, café, or flower store (Landesco 1929). For those engaged in sex trafficking, particular cafés, clubs, and cabarets are important as well. Snow and Johnson say that there is an underworld in every large city anywhere in the world and, within it, places where the traffic in women takes place. At these locations, *souteneurs* discovered opportunities and threats, learned the movements of their associates, and obtained money or other assistance to transport women (League of Nations 1927a, p. 24).

Everyone agrees that the primary motive for organized crime is financial gain. Landesco emphasized illegal markets created by legal prohibitions. Illegal enterprises seek to create monopolies in particular industries as did legitimate businesses (Landesco 1929, p. 1093). This insight has come to be known as "illegal enterprise theory" (Haller 1990; Liddick 1999; Reuter 1983). Snow and Johnson regarded the traffic in women as a commercial enterprise governed by the laws of supply and demand. "We have used these economic terms because they seem aptly to describe the commercial aspect of the whole traffic. There exists or arises, owing to a variety of causes, a demand for prostitutes in some particular area. The trafficker sets out to supply it" (League of Nations 1927a, p. 9). In addition to licensed houses in particular cities, trafficking followed other economic conditions: areas with a surplus of men or women; areas with seasonal movements of men (sailors, soldiers, or tourists); theaters, clubs, and cabarets that encouraged prostitution (League of Nations 1927a).

However, in thinking about crime in economic terms, disagreement emerges. In the Chicago-based theory, commercial enterprise requires bureaucracy. Organized crime operations are analogous to ordinary businesses in that they exist to make money but also because they share a similar management structure. Landesco uses "trust", "syndicate", and "ring" to describe the control of prostitution in Chicago (Landesco 1929, pp. 845–153). Thrasher defined a ring as "a permanent conspiracy usually made of 'inside men' associated secretly for some illegitimate purpose profitable to themselves" (Thrasher 1927, p. 437). This sounds like what Vollmer was talking about in Los Angeles. Landesco says that organized crime "had no formal organization" but was rather a "gangster form of organization" held together by its leaders. Professional criminals forged alliances, loyalties,

and partnerships with corrupt police and politicians (Landesco 1929, p. 1092). It also included individuals in particular roles. One of these was that of "fixer", that is, police, lawyers, or politicians who could alter the course of criminal proceedings (Landesco 1929, p. 1057; 1934, pp. 346–47). Presumably, Judge Craig and Judge Richardson would have had this role in the Los Angeles ring, as well as the lawyer, Warren Williams.

For Snow and Johnson, there are no rings or syndicates. "The real underworld rings, so far as they exist, are a different affair from that usually imagined" (League of Nations 1927a, p. 6). The undercover investigators had made contact with "certain prominent characters" in the underworld. In this way, they managed to "penetrate to the center of the so-called ring" and traced the traffic from country to country by "securing of introductions to the 'right people' in each center studied". Snow and Johnson emphasized there was no organized international group, but a "worldwide camaraderie", held together by overlapping financial interests, shared knowledge of evading legal restrictions, and a common jargon of commercial sex. "No formal organization exists to further the traffic, but there is evidence of local associations of traffickers in many different countries with groups of their own and meeting places" (League of Nations 1927a, p. 24).

Snow and Johnson do not see traffickers in competition. They do not provide examples of violence to build syndicates or trusts, nor for that matter, to control women. Rather, the traffic in women involves individuals engaged in many activities who cooperate to advance their interests. Networks developed for one illegal commodity overlapped with others, whether illegal alcohol or illicit drugs (League of Nations 1927a, p. 9). Snow and Johnson identified four roles. (1) The principals are usually the owners of brothels, often retired *souteneurs* who provide funds and gather information to coordinate the traffic. (2) *Madames* managed brothels. They tried to recruit new women, often in association with principles and *souteneurs*, who found ways to keep women indebted to the brothel. (3) *Souteneurs* financed the business, secured new girls, ferried them to brothels, and introduced them to the *madames*. They were constantly travelling. (4) Finally, intermediaries transported women for the *souteneurs* and *madames*, recruited artistes and entertainers, found customers for brothels, and liaised with sailors who brought new women (League of Nations 1927a).

The concept of "trafficking" emerges from the League's investigation. Snow and Johnson discarded the "legalistic definition" of traffic and formed their own working definition. The legal definition was too narrow in that it limited traffic to cases of women under twenty-two years of age or if over that age, procured by force or fraud (Johnson 1928, p. 68). For their study, they regarded "international traffic" as "direct or indirect procuration and transportation for gain to a foreign country of women or girls for the sexual gratification of one or more other persons". It included employment agents obtaining women as entertainers and artistes and profiting from their prostitution (League of Nations 1927a, p. 9). This was an important insight because it meant they could explain international trafficking without relying on professional criminality. Rachel Crowdy made clear in 1927 that the investigators had explored the possibility of "what one might call a ring with a super-trafficker sitting at the head making profits out of a big organisation" and "found nothing of the kind" (Crowdy 1927, p. 157).

As Snow and Johnson explained, *trafficking* could take place without *traffickers*. In the portion of the final report devoted to the United States, they gave an example of the traffic in girls from New York to Panama. A young woman argued with her roommate, lost access to her apartment, and decided she wanted to get away from New York City. She met a theatrical agent who offered her a new life as a jazz singer at a cabaret in Panama and told her she could make extra money by having sex with Navy officers. Having arrived in Panama, it was not what she expected, and she wanted to return home. However, she could not afford the return fare until completing her contract. The proprietor of the cabaret did not allow the women to use rooms at his club but encouraged them to drink with customers. The proprietor had an understanding with the theatrical agent about sending girls willing to drink with customers and spend the night with them afterward. The agent makes money by sending girls willing to engage in casual sex, and the proprietor makes money from the

additional customers attracted by the presence of sexually available females. The young woman turns to prostitution and eventually agrees to a second contract. As Snow and Johnson explained, most of the cabarets overseas amounted to little more than houses of prostitution, and the recruiting of American girls for them under the guise of entertainers and artistes "is international traffic" (League of Nations 1927b, p. 12).

In this example, Snow and Johnson have explained how the traffic in women should be understood as a "social network" or "criminal network" (Kleemans and van de Bunt 1999; Morselli 2009; Carrington 2011). The individuals share a common interest in financial gain which leads them into particular alliances. Each person—the theatrical agent, the cabaret proprietor, and even the jazz singer herself, make decisions to advance their own interests, all of which add up to her victimization. The network produces traffic in women, yet there is no criminal gang, nor even a professional criminal, involved. Trafficking in women in other contexts involved more intermediaries, increased transactions, a wider network, and even a procurer. Yet it is the same system, and that system will be extended to other commodities, such as drugs, making it a theory of organized crime. Where Vollmer looked for the spider, Snow and Johnson looked at the web.

Although both Vollmer and Snow and Johnson are trying to understand organized crime, they start with different problems. For Vollmer, the breakdown of criminal justice is the problem. Los Angeles afforded an example of the "hold that vice has upon the government" (Vollmer 1936, pp. 83–84). The ring provides a suitable concept because it explains how criminals, working with corrupt politicians and police, carry on their activities without prosecution. In Los Angeles, Vollmer perceived the criminals to be more organized than the police. He found the same thing in Chicago, only on a larger scale (Vollmer 1936, pp. 83–84). Landesco made the same observation. To explain it, he dug deep into urban organization and links forged over time in the micro-societies of neighborhoods. He found alliances between gangsters, police, and politicians that went back decades before Prohibition. These relationships remained consistent while criminal justice remained inconsistent. Politicians, police leadership, and reformers came and went while corrupt forces stayed put (Landesco 1929, pp. 1092–93).

Snow and Johnson had a different problem to solve, namely, how to explain an inter-continental system that moved women on an industrial scale without any bureaucratic structure. They were not looking for rings or ringleaders, or trying to bring individuals to justice, but rather, trying to destroy a wide-scale system of exploitation. Trafficking would be abolished, not by making arrests, but by extending regulation. The system of toleration, or licensed houses in certain cities, created a demand for "fresh attractions". Trafficking in women represented the supply. Essentially, the laws to abolish prostitution in the United States should be extended to other countries. Closing down licensed houses in Buenos Aires, Cairo, etc., would undermine the demand, the market for women would diminish, and the "sinister business" of trafficking in women would slide into a terminal recession.

Snow and Johnson may not have come up with the ultimate explanation for sex trafficking. They simply projected the American plan onto the rest of the world, assuming that what was true for American cities was true in every other city. They brought pre-conceived ideas to their research, as well as their own racial and sexual prejudices (Schettini 2017; Robertson 2009). Yet, in developing their explanation of sex trafficking, Snow and Johnson advanced the theory of organized crime, particularly when compared to police projects known to the League of Nations. These projects travelled the road of international supervillains leading multinational criminal corporations.

In 1933, the International Criminal Police Commission attempted to define "international criminal" and "international crime". Their discussion began with a definition formulated in 1905 at the Union Internationale de Droit Penal, which essentially said that any crime could be considered an international crime if any element of it (planned, facilitated, or carried out) had an effect on more than one country. The discussion focused, however, on the hunt for "professional criminals". Dr. Antonio Pizzuto, Italian Federal Police, insisted that international crime could not be understood apart from understanding

international criminals. It was necessary to "control and register the criminals of any international kind". There were two fundamental elements distinguishing international criminality from ordinary criminality. First, the intention of the perpetrator, which had become a habit; and second, the field of action chosen by the perpetrator, that is, the effort to escape from a criminal penalty (ICPC 1933).

The other project took place within the other technical organization authorized by Article 23c of the Constitution, the Advisory Commission on Opium and Other Dangerous Drugs, also known as the Opium Advisory Commission (OAC). The OAC received an introduction to drug smuggling from T.W. Russell Pasha, head of the Central Narcotics Bureau of Egypt. From information obtained from smugglers, he learned the routes into Egypt and traced these back to clandestine laboratories in Europe. In his annual reports, sent to leading newspapers as well as the OAC, he portrayed drug smugglers as clever individuals who devised ingenious methods of sneaking their illicit products past the authorities. He presents drug trafficking as an illegal enterprise with a bureaucratic structure. His first report narrates how he smashed the Zakarian organization. Thomas Zakarian had started from a small shop in Alexandria selling carpets and became an international drug dealer. Zakarian went into partnership with an international ring led by the Zelinger brothers in Vienna, and they built smuggling routes back to Egypt from their clandestine laboratory in Switzerland (Central Narcotics Intelligence Bureau 1930, pp. 1–8).

### 5. Conclusions

To make sense of August Vollmer's report of a "white slave traffic ring" in Los Angeles in the 1920s, it is useful to think of him in a way we are not used to thinking of him: as a criminological theorist. Most of the names on his list were already known criminals, so the significance is not in exposing a hidden aspect of Los Angeles at the time, but rather, in how to interpret the organization of criminal activity. What he says about prostitution in the city offers a rough sketch of the more detailed picture of organized crime being drawn in Chicago at the time. Frederic Thrasher and John Landesco use words such as "ring", "syndicate", and "trust" to explain how gangsters conduct their illegal activities and manipulate the criminal justice system. However, thinking about organized crime in this way affirms the Chicago-based theory as an explanation best-suited for analysis of urban areas and local markets for illegal products and services (Yeager 2012). Claims about white slave gangs operating on an international scale, reported in the press, were found to be false. Investigations repeatedly failed to turn up evidence of such an organization.

In preparing their report for the League of Nations, William Snow and Bascom Johnson discounted Vollmer's report because it did not fit the theory of trafficking in women they had developed. "Trafficking theory" agreed that traffic in women was an illegal enterprise best viewed in economic terms, but discarded assumptions about managerial control. They did not see rings, syndicates, or trusts at work. Rather, they perceived a vast system of alliances, relationships, expectations, and agreements among individuals, few of whom could be considered professional criminals. In this way, they provided an early version of social network theory. This allowed Snow and Johnson to explain trafficking without traffickers and to see organized crime without conventional ideas of organization, whether it be professional criminality or bureaucratic control.

Taking the League's work into account adds to the theory of organized crime. Emerging from the 1920s is a wider stock of concepts that can accommodate both local and international activities. The word *trafficking* has become ubiquitous in discussions of global crime. There are references to trafficking not only in drugs but a wide variety of illegal commodities from exotic animals to nuclear weapons. An important question to ask is how far the work of Snow and Johnson can be extended to other criminal activities, particularly global crime. Additionally, an important way to find out is to explore the activities of the League of Nations within the wider history of organized crime. Some historical studies have appeared, but serious work on the League of Nations and its legacy for criminology has only just started.

**Funding:** This research received no external funding.

**Conflicts of Interest:** The author declares no conflict of interest.

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
