# Peer review of "August Vollmer, Traffic in Women, and the Theory of Organized Crime"

_socsci, doi:10.3390/socsci11070283_

Round 1

Reviewer 1 Report

Review August Vollmer, Traffic in Women, and the Theory of Organized Crime

This is a well written article that tells an interesting and accessible story of exchanges over white slavery, trafficking and prostitution. While the article sheds light on important dynamics in how organized crime has been understood and on the relationship between local, national and supranational institutions and understandings, its contribution to criminological theory is not evident. The authors argue that Vollmer can be seen as someone who contributed to thinking about organized crime, but the way he worked and the insights that he presented, might as well be seen as linked to the development of naturalistic news reporting that grew out of Chicago at the same time. He was seemingly behaving more like an undercover reporter than a stringent scholar. The main bulk of the article is organized in three sections, one describing the activities of the League of Nations in California, one describing Vollmer’s work and connections in Los Angeles, and one describing developments in theories of organized crime. The two first sections are highly descriptive and consists of details that are interesting, but much of it is not really relevant for the argument of the article. The two sections are also not clearly connected to the third section, outside the fact that Vollmer on the one hand, and Snow and Johnson on the other, subscribed to different understandings of organized crime. I would like to see section three, which now presents an interesting overview of early conceptualizations of organized crime, expanded and to more clearly demonstrate Vollmer as someone who contributed to criminology. Now it is just stated several times that Vollmer adds to theory of organized crime more than it is demonstrated.

Also, the article lacks an overview over how the material has been produced, what it consists of and how identified material has been interpreted. This is needed for this to be published as an article, in my opinion.

Author Response

Thanks for reviewing my paper.  I appreciate your comments, especially about the need for an explanation of methodology.  I've added a paragraph to the introduction that explains the method and theory of historical criminology.  Your suggestion about expanding the discussion of Vollmer is more complicated because his account a white slave traffic ring in L.A. is meaningful only in the context of the League of Nations' worldwide investigation; and specifically, in contrast to Landesco's work in Chicago and Snow and Johnson's work in port cities.  So the third part makes sense only after the first two parts.  Otherwise, Vollmer insists on a dramatic conspiracy version of the white slave trade than no one doing research/invesigation supported. 

Reviewer 2 Report

Methodology-wise, little is being said about the material used beyond being in the League of Nations archives at the UN in Geneva. Even though the scope of the article is more humanities-based and a full sociology methods section is not needed, maybe the author(s) might want to share some more insight into the research process. 

Author Response

Thank you for your review.  You are quite right, the paper needs more explanation of methodology.  I've added this to the introduction--a discussion of historical criminology as a method of inquiry, along with cites to others working in this area that provide examples.

Reviewer 3 Report

I have no comments or suggestions.

I accept the article in this form, treating the details as the licentia poetica of the Author. 

Author Response

Thank you for your review.  I have tried to improve the paper by adding an explanation of methodology--historical criminology.

Round 2

Reviewer 1 Report

I see the authors' point on not expanding section three, so I will not repeat that comment, but I still feel more can be said about what the material consists of and how the authors have analysed it. The authors have revised the piece to frame it clearly as historical criminology, not not much more has been added on how that has informed how the authors have worked with the sources.